# "There is no reason anybody should be using 1D anymore": Design and Evaluation of 2D Jupyter Notebooks

Jesse Harden[*]
Virginia Tech

Elizabeth Christman[†]
Virginia Tech

Nurit Kirshenbaum[‡]
University of Hawaii at Manoa

Mahdi Belcaid[§]
University of Hawaii at Manoa

Jason Leigh[¶]
University of Hawaii at Manoa

Chris North[‖]
Virginia Tech

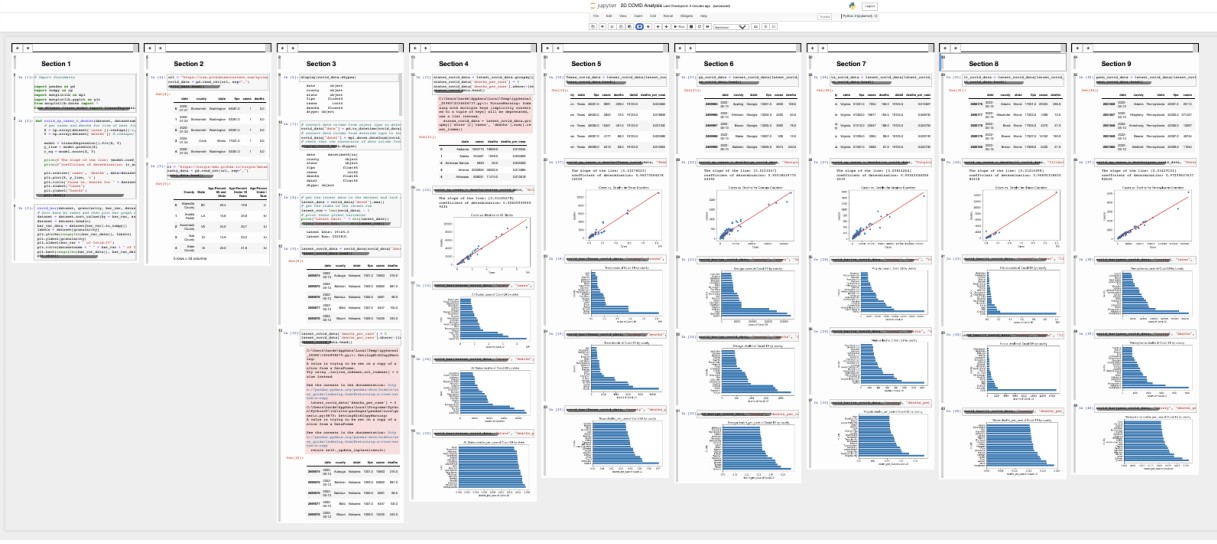

Figure 1: Finding & Comparing Results 2D Notebook from Study 1

## ABSTRACT

Current computational notebooks, such as Jupyter, are a popular tool for data science and analysis. However, they use a 1D list structure for cells that introduces and exacerbates user issues, such as messiness, tedious navigation, inefficient use of large screen space, performance of non-linear analyses, and presentation of non-linear narratives. To ameliorate these issues, we designed a prototype extension for Jupyter Notebooks that enables 2D organization of computational notebook cells into multiple columns. In this paper, we present two evaluative studies to determine whether such "2D computational notebooks" provide advantages over the current computational notebook structure. From these studies, we found empirical evidence that our multi-column 2D computational notebooks provide enhanced efficiency and usability. We also gathered design feedback which may inform future works. Overall, the prototype was positively received, with some users expressing a clear preference for 2D computational notebooks even at this early stage of development.

**Index Terms:** Human-centered computing—Human Computer Interaction (HCI); Human-centered computing—Visualization

---

[*]e-mail: jessemh@vt.edu

[†]e-mail: elizabethc99@vt.edu

[‡]e-mail: nuritk@hawaii.edu

[§]e-mail: mahdi@hawaii.edu

[¶]e-mail: leighj@hawaii.edu

[‖]e-mail: north@vt.edu

## 1 INTRODUCTION

Computational notebooks like Jupyter [18, 24], used to construct and present computational narratives [35, 42, 45], struggle with non-linear analyses, such as comparative analyses, and non-linear narratives [13, 42], as well as navigating longer notebooks [13], preventing and managing messiness [10, 14, 21, 28, 29, 42], and efficiently using large display spaces [13]. We suggest that part of the reason for these issues is the current 1D, top-to-bottom organization of notebook cells.

Weinman et al.'s work on Fork-It [49] showed 2D space can be helpful; they introduced forking, the temporary creation of split columns in an otherwise 1D notebook. While this work helps non-linear analyses, it does not easily accommodate non-linear narratives, which may benefit from a persistent multiple column approach. Wang, Dai, and Edwards [48] also sought to shift computational notebooks from the current 1D structure with Stickyland, which allows users to "stick" cells to a dock that is persistently at the top of the computational notebook interface even when scrolling. Harden et al. [13] explored how users would arrange cells in 2D and found three different patterns: linear (with either split cells or split columns), multi-column, and workboard. This work demonstrated alternative organizations of cells, some of which would not be possible in the prior works mentioned; it also suggests that computational notebook users could benefit from 2D space usage for organizing notebook cells in a more flexible yet persistent manner.

This paper contributes to computational notebook research through evaluations of a 2D layout extension for computational notebooks. We focused on the following research questions:

1. When comparing 1D and 2D layouts, which mode supports more efficient user completion of data science tasks, such as

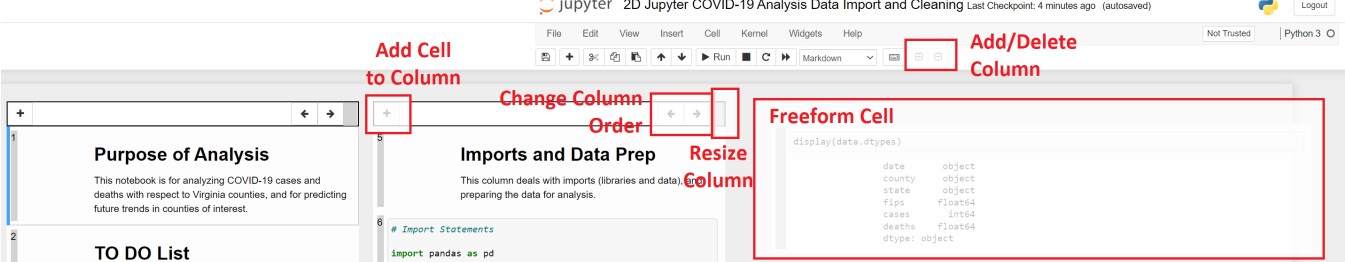

Figure 2: Notebook Controls for 2D Jupyter extension

information retrieval, results comparison, parameter tuning, and code comparison?

2. What strengths & weaknesses might 2D have compared to 1D?

3. Would users find 2D layouts more usable than 1D layouts?

4. Would users prefer to use 2D for computational notebook cells?

To answer these questions, we designed a Jupyter Notebook extension that enables a 2D multi-column cell layout. We then conducted two user studies using this extension where users performed a series of tasks in both 1D and 2D layouts, followed by qualitative data gathering through surveys and, in the second study, interviews. The first study used pre-made notebooks to evaluate whether the extension enhances performance and usability, while the second study focused on creation of a 2D notebook from scratch for a data science task. We found 2D layouts provided more efficient user task performance and enhanced usability over 1D layouts. Users overwhelmingly preferred the 2D notebooks, and made use of available display space to organize notebooks such that more cells are simultaneously visible. We also noted some design challenges for 2D layouts, including managing column width in a multi-column layout.

## 2 BACKGROUND AND RELATED WORKS

This work builds on two key areas of research: computational notebooks and Space to Think.

### 2.1 Computational Notebooks

Computational notebooks support incremental and iterative analysis [19, 42] and computational narrative formation through interleaving code, visualizations, and text [34, 42]. However, computational notebook users face various issues and pain points [6], such as messiness [14, 22, 29, 42], dealing with non-linear analyses and narratives [42], and navigating longer notebooks [13]. These issues may be exacerbated by the current 1D structure of computational notebooks.

Head et al. [14] showed messiness can come from disorder, deletion, and dispersal, where disorder means run order and presentation order are different, deletion means overwriting or deleting necessary code, and dispersal means related cells are far apart. Many tools have been developed to help deal with messiness, from Head et al.'s work [14], to cell dependency graph visualization [50] to version control systems for computational notebooks [20, 21]. The 1D structure may exacerbate messiness given the looping nature of sensemaking in computational notebooks [37, 38], so 2D space usage may help minimize it.

Scrolling through a long notebook can be tedious and negatively affect various tasks like debugging and cleaning. While Google Colaboratory [12] enables jumping to different sections through a table of contents, the 1D structure can still result in tedious scrolling.

Exploration of 2D space usage by Weinman et al. [49] and Harden et al. [13] produced positive responses. Within the bounded 2D of

Fork-It [49], users did more than just comparative analyses; they used the split column structure to organize code and contain messes. Harden et al.'s [13] findings corroborate these potential use cases.

#### 2.1.1 Computational Notebooks & Reproducible Science

Reproducible research is an important and challenging issue for any scientific endeavor, and research in HCI and computer science is no different [3, 7, 11]. At their best, computational notebooks and the computational narratives formed using them enable reproducible scientific workflows [2, 24, 41]; the ability to interleave documentation with code and results contributes to this potential. However, computational notebooks in the wild are rarely reproducible [36]; issues such as messiness [14], out-of-order execution [36], and dependency issues contribute to the pain of trying to reproduce computational notebook findings [6]. Indeed, less than 5% of notebooks studied by Pimentel et al. [36] were reproduced with the same results. Work to address issues with reproducibility, such as Osiris by Wang et al. [47], has helped; while our work does not directly focus on reproducibility, its ability to enable expanded use of space may indirectly help alleviate issues affecting reproducibility.

### 2.2 Space to Think

Andrews et al. [1] found large, high-resolution displays benefit sensemaking in 2 key ways through what they called "Space to Think": external memory and semantic encoding. External memory means more information can be stored on screen space instead of in one's mind, which allows physical navigation, like moving one's head, to replace virtual navigation, like scrolling or changing tabs. Semantic encoding means users can group related items spatially based on their mental model of the connection between items; in short, users can externalize their understanding onto the screen. Recent studies [8, 26, 27] have expanded this concept to the space provided by virtual and augmented reality or cited Space to Think as an influence on their design [33, 39, 40]. Kirshenbaum et al. [23] found Space to Think can also benefit collaborative meetings.

Current computational notebook systems with their 1D structures do not adequately use Space to Think without clumsy workarounds like opening the same notebook multiple times and arranging side-by-side. 2D space usage may enable Space to Think in data science tasks [13]. To this end, some recent tools, such as VisSnippets [5], Einblick [16, 44], CoCalc [17, 31, 46], and Code Bubbles [4], have begun to explore 2D layouts of cells using a whiteboard metaphor.

## 3 DESIGN OF 2D JUPYTER NOTEBOOK EXTENSION

Harden et al. [13] found two main categories of 2D layouts for computational notebooks based on user-generated layouts: multi-column and workboard, both of which are supported by the **2D Jupyter** extension we developed and evaluated; the extension and supplemental materials for this paper can be found at https://github.com/infovis-vt/2D-Jupyter on GitHub. Multi-column is fully supported. Workboard, or more complex structures such as directed graphs and nested columns and rows, is enabled by freeform dragging of cells. Given

that the multi-column pattern in Harden et al. [13] was consistent across all its constructions and frequently constructed (32% of all participants), in addition to being part of several of the grouped combinations workboard sub-pattern, it makes sense to focus on enabling and evaluating the multi-column pattern first.

To support multi-column layouts, 2D Jupyter enables creation and deletion of columns, resizing and re-ordering of columns, adding cells to a column, and moving cells from one column to another; This is done through user interface (UI) additions, as seen in Figure 2. The Plus and Minus buttons on the main toolbar create and delete individual columns respectively. Also, each column now has a toolbar at its top; The bold Plus button here adds a cell to the column, the left and right arrows reorders the column in the arrow direction, and the gray box can be clicked and dragged to resize a column. In addition, cells can be dragged to another column by clicking and holding the new gray box on each cell's left side. Finally, the Run All functionality is preserved in a top-down, left-to-right format; in other words, the leftmost column's cells are run in top-down order, followed by the cells in the column immediately to the right, repeating until all cells are run.

To enable workboard layouts, each cell can be dragged and placed outside of the columns, as seen in the freeform cell in Figure 2. More advanced workboard features, such as arrows to connect cells or other whiteboard annotations are not yet implemented. In addition, cells outside of columns are not run as part of the Run All functionality. For now, we suggest using workboard freeform cells for more ephemeral uses such as scratch space, viewing data, and other tasks not relevant to the final computational narrative.

## 4 STUDY 1 METHODOLOGY

The goal of our first study is to measure and compare user task performance in 1D and 2D notebooks. We therefore conducted a controlled study consisting of a pre-screening questionnaire, a set of user performance tasks, and survey questions. The study design had one within-subjects variable, *layout* with two treatments, *1D* and *2D*; and one between-subjects variable, *order* with two treatments, *1D-first* and *2D-first*. The user tasks focused on research question 1; participants completed three task sections in both 1D and 2D. For the surveys, we focused on research questions 2-4.

### 4.1 Recruitment and Screening

89 potential participants, recruited via academic listservs of students and faculty from a large state university, responded to an online screening questionnaire asking whether they had experience with both Python and computational notebooks such as Jupyter. 62 potential participants passed the screening due to having experience with both Python and computational notebooks and were notified that they could take part in the study. Of these 62, 31 chose to take part in the study. We discarded 1 of these 31 participants' data due to technical issues that arose during the study, leaving 30 participants. 15 of these participants were assigned to 1D-First, while the other 15 were assigned to 2D-First. Participants were randomly assigned to a group, with the only restriction being balancing the group numbers so that they were as equal as possible.

### 4.2 Hardware for User Study

For the user study tasks, participants used an iMac computer with a 24-inch monitor and either an iMac mouse with a built-in trackpad for horizontal and vertical scrolling, or an external trackpad with horizontal and vertical scrolling that also had buttons for clicking. The monitor was wide enough to display 4 to 5 columns of the notebook at a time.

### 4.3 Task Designs & Rationales

The tasks were designed to mimic common data science scenarios performed in computational notebooks. We created 6 computational notebooks (3 1D, 3 2D) for this study; the 2D notebooks used the multi-column pattern due to it being the most common, consistent pattern seen in Harden et al. [13] as mentioned in Section 3 and being fully supported by our extension. Each notebook was designed for one of three task sets: Finding & Comparing Results, Parameter Tuning, and Code Comparison. Each layout (1D, 2D) and task set combo had one notebook, and each task set's notebooks were slightly different so participants could not memorize answers between layouts. However, the differences were designed to not impact difficulty between the tasks in 1D and 2D. Users had the notebooks open, one at a time, on the iMac, while the user study survey, with questions and instructions, was open on a separate laptop. Users were allowed to use any existing functionalities, such as searching for information using Control + F; we did not suggest such methods unless a participant asked.

To compare 1D vs. 2D, we measured the time it took a participant to answer the survey question and press the "Next" button on the survey as time to completion, and accuracy for all tasks was a count of correct answers; we also measured the number of times and amount of time spent scrolling for the code comparison task. 16 participants started with the 1D notebooks first, and 15 participants started with 2D first; This design, along with training in the first notebook layout type for each person, helped counterbalance the study to minimize bias from repeated tasks. One 1D First participant's data was discarded due to technical issues. Each participant took at most 1 hour to complete the study.

The 6 notebooks and a copy of the study session survey can be found at https://github.com/infovis-vt/2D-Jupyter on GitHub.

#### 4.3.1 Finding & Comparing Results Task

Harden et al. [13] found that users expected finding and comparing tasks to be better in 2D layouts than in 1D layouts. Thus, this task set sought to measure statistically whether such a benefit exists.

The notebooks for this task set contained COVID-19 data analysis for the USA by state and then for 5 individual states by county, as seen in Figure 1. Sections 1-3 of these notebooks had cells for imports, function definitions, and data preparation, while Sections 4-9 had cells that analyzed and visualized results for each geographic region as a scatterplot and 3 bar charts. In data science, such notebooks often result from copying-and-pasting cells for parallel analyses of different data subsets. The 1D notebook design concatenated these sections into a single long list of cells. In the 2D notebook, each of the 9 sections was separated into its own column of cells, with columns arrange left to right. This notebook design was based on common layout strategies previously observed by Harden et al. [13], where a common strategy was to organize parallel analyses in side-by-side columns to enable easy comparison.

For this task set, we included a find task, a graph comparison task, and a numerical comparison task. We did not allow participants to look over the notebook before beginning the task set.

In the find task, participants had to locate info in the notebook based on the notebook structure. The question was of the form *"Which state's analysis is found between the analysis of STATE1 data and the analysis of STATE2 data?"* We measured the time it took each participant to retrieve the info in 1D vs. 2D notebook layouts. The hypothesis was that spatial 2D columns would enable more rapid recognition and access to relevant notebook sections.

In the graph comparison task, participants had to compare results in several different charts throughout the notebook. The question was of the form *"Out of those shown in the relevant bar charts, which county in which state, EXCLUDING the ALL STATES section, had the highest number for ATTRIBUTE of COVID-19?"* We measured the time it took each participant to compare charts in 1D vs. 2D notebook layouts. The hypotheses was that 2D column structure that aligned parallel analyses would enable faster comparison by horizontally scrolling through the corresponding charts, whereas

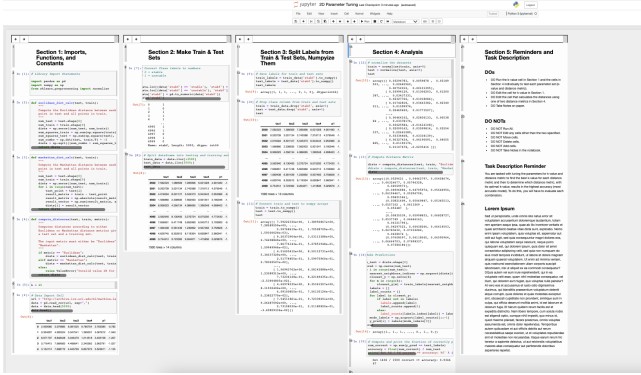

Figure 3: Parameter Tuning 2D Notebook

the 1D notebook would require significant vertical scrolling and searching for each chart to compare.

Similarly, in the numerical comparison task, participants were asked a question of the form *"Which section's scatterplot graph's line of best fit least/best fits the data (coefficient of determination closest to 0/1)?"* The coefficient of determination was displayed above each scatterplot. We measured the time it took each participant to compare numerical results in 1D vs. 2D notebook layouts.

### 4.3.2 Parameter Tuning Task

A common problem in data science involves testing various parameter values for an ML model. The notebooks for this task, as seen in Figure 3, contained K-Nearest Neighbors (KNN) algorithm used to analyze network stability data. Participants were instructed the following: "You will be asked questions that require tuning the parameter 'k' in Section 1 and choosing the distance metric in Section 4. Only run the necessary cells (the "k-value" cell in Section 1, and the cells in Section 4) to test each possible parameter set (k-value and distance metric)." In each notebook, the cell which assigns the k-value was in the first section while the code for calculating the distances, making predictions, and determining accuracy on the test set were in the fourth section; participants were not allowed to move cells. Participants were asked three questions in the following order, with different k-value options for 1D and 2D:

1. Which of the following k-values produces the most accurate model with the given dataset for the Euclidean distance metric?

2. Which of the following k-values produces the most accurate model with the given dataset for the Manhattan distance metric?

3. Given each distance metric with its optimal k-value, which distance metric produces the most accurate model on the given dataset?

In data science endeavors, code near the beginning of a notebook can influence results later on in the notebook; While it is possible to move such dispersed cells closer to each other, such re-ordering is not always feasible depending on the design of the analysis. Thus, we sought to simulate a situation in which one wants to retain the given order while continuing their analysis. The goal here is to see if 2D notebooks, with a layout where each section has its own column, can minimize the effects of dispersal [14] by making cells that are far apart in a 1D layout effectively closer on the screen in a 2D layout and lead to performance improvements. Thus, we measured how long it took participants to answer all three questions together.

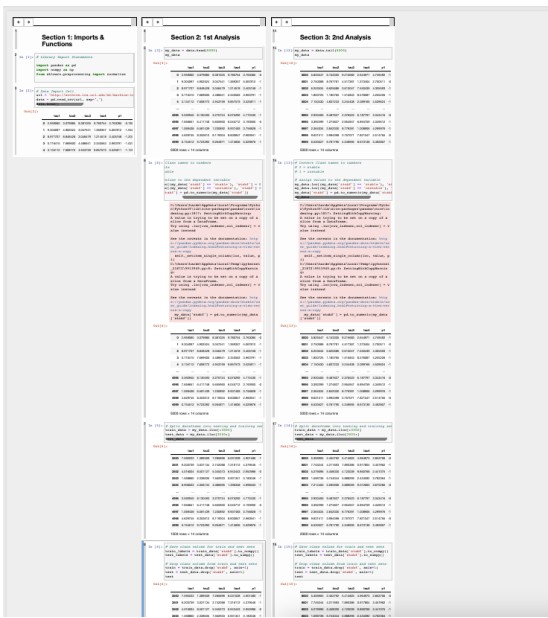

Figure 4: Code Comparison 2D Notebook Clip

### 4.3.3 Code Comparison Task

Data scientists often need to compare the code for multiple versions of a model to understand differences. The notebooks for this task, as seen in the right image in Figure 4, contained two runs of a K-Nearest Neighbors ML algorithm with several code differences between them. Participants had to choose which items from the list of options, ordered in terms of appearance, differed between each run. The 2D notebook organized the two runs into adjacent columns. The list of differences included items such as the following:

1. The cutoff number for the training and testing splits

2. Different distance metrics (Manhattan, Euclidean) used

3. The variable name for the distance matrix

4. The value of k (number of nearest neighbors)

The goal of this task was to test how quickly users can find differences between two similar sets of code, which often happens when debugging model errors. Given that Harden et al. [13] found significant skepticism about the potential of 2D notebook layouts for debugging, it makes sense to test this important debugging sub-task.

### 4.4 Survey Questions Design

Likert-scale questions were used at the end of both the 1D and 2D task sections, and after both sections were completed. The 5 questions at the end of the 1D and 2D task sections focused on rating each layout individually, without comparison to the other, while the 13 questions at the end focused on comparing 1D and 2D layouts; these 13 questions were largely taken from Harden et al.'s experiment [13]. After the 13 questions was a comment box where users could elaborate on any answers they gave.

The questions after each of the 1D and 2D task sections focused on perceptions of usability for the layout on the given tasks; We compared their answers between layouts to better understand whether users saw potential improvements in 2D layouts over 1D layouts.

Table 1: P-values for Scheirer-Ray-Hare by Task and Effect

| Task | Order | Layout | Interaction | Mean Improvement by 2D | Median Improvement by 2D |
|---|---|---|---|---|---|
| Find | 0.137 | 0.054 | 0.594 | N/A | N/A |
| Graph Comparison | 0.255 | **4.2e-5** | 0.131 | **32%** | **45%** |
| Number Comparison | 0.559 | **5.8e-6** | 0.156 | **46%** | **34%** |
| Parameter Tuning | 0.779 | **0.003** | 0.487 | **19%** | **23%** |
| Code Comparison | 0.882 | **3.5e-4** | **4.6e-4** | **34%** | **33%** |

Bolded values are statistically significant with a 0.05 threshold. All other values are not statistically significant.

## 4.5 Data Analysis Process

We divided the quantitative data analysis for Study 1 into 3 areas: Efficiency Measurements, Survey Questions, and Scrolling Time.

### 4.5.1 Efficiency Measurements

We started our analysis of efficiency using 2-Factor ANOVA. However, due to completion time being log-normally distributed and thus violating the assumption of normality for ANOVA, we also used the Scheirer-Ray-Hare Test [43], a non-parametric alternative to 2-Factor ANOVA and extension of the Kruskal-Wallis Test [25], to test if layout (1D or 2D), as well as order (1D First, 2D First) and interaction between layout and order, affected time to completion; significant results were followed up with calculations of the differences between the means and medians of the 1D and 2D layouts to determine whether the 2D layout resulted in more efficient performance and to calculate average time saved as a percentage. R [15] and the packages RCompanion [30] and FSA [32] were the main tools in this analysis.

Accuracy was measured by counting the number of questions answered correctly in each layout by all participants and then dividing by the multiplication of the number of participants and the number of questions.

### 4.5.2 Survey Questions

For the Post-1D and Post-2D questions, we created and analyzed a bar chart of average rating by order and layout, and a heatmap of ratings. We also tested the statistical significance of the differences in ratings using a paired t-test and the Wilcoxon test, inspired by work by De Winter and Doduo on analyzing Likert-Scale questions [9]. For the Post-Experiment questions, we made and analyzed a heatmap of ratings. The qualitative comments were analyzed for themes using open coding by the first author. After the initial pass, feedback from the other authors was sought on the themes and used to refine them. Then, a second pass was made with all quotes grouped into themes.

### 4.5.3 Scrolling Time

To determine the amount of scrolling done in 1D vs 2D, we recorded scrolling events, including the time taken to scroll, while watching the footage for each participant's Code Comparison task work in 1D and 2D. We limited events to scrolls for navigation as opposed to micro-scrolling events that do not bring new cells into view; we did this by only considering those scrolling events that lasted for at least 2 seconds. To determine scrolling endpoints, we looked for breaks between scrolls lasting at least 2 seconds; scrolling events with smaller breaks than 2 seconds were considered as 1 event for the purpose of this analysis.

## 5 STUDY 1 RESULTS

We divide our results into 4 areas: User Interaction Strategies, Efficiency Measurements, Survey Questions, and Scrolling Time.

## 5.1 User Interaction Strategies

Our observations of user behaviors with the 1D and 2D layouts, divided by task notebook, are summarized here.

### 5.1.1 Finding & Comparing Results Task

In 1D, all users started by scrolling down through the notebook to answer the Find question (which state's section was between two other states' sections) until they found the answer. Then, they scrolled through Sections 5 through 9 to answer the graphical Comparison question (which county in which state had the highest value for a particular variable) and compared the bar chart results and axes, which was sufficient to find the highest value. Some users, because they forgot a previous value or wanted to verify their memory, would scroll back to earlier results, sometimes multiple times, before submitting an answer. A couple users took notes on paper to avoid this issue. For the numerical comparison question, users repeated the process for first Comparison question with Sections 4 through 9.

In 2D, all users started by scrolling to the right to answer the Find question. Since the columns for the relevant sections (4-9) were fairly well aligned, as seen in Figure 1, this mitigated the need to perform vertical scrolling except for between questions. Users scrolled less distance in 2D due to more efficient use of space with 1 column representing 1 section. Then, to answer the 2 Comparison questions, all users used physical navigation (e.g. head movement) with less scrolling needed, since the screen could show 4 columns. The efficient, well-organized use of 2D also led users to perform less backtracking, if any, and eliminated the need to take notes on paper.

### 5.1.2 Parameter Tuning Task

In 1D, all users repeatedly scrolled up and down to get results for different parameter combinations (k-value and distance metric). Sometimes users scrolled past the cells they were looking for and thus did additional scrolling to correct their focus. All users took notes on paper so they could remember and compare results.

In 2D, much smaller scrolls were needed to get from the first column, where the main parameter was, and the fourth column, where results were calculated. Given the much smaller scrolling distance, scrolls were quicker and did not result in scrolling too far nearly as often. All users also took notes on paper with 2D, as well.

### 5.1.3 Code Comparison Task

In 1D, all users scrolled up and down to find code differences in the two different analyses; users examined the code in a cell in the first analysis, then scrolled down to examine the code in the corresponding cell in the second analysis before scrolling back up again to look at the next cell. This process was repeated until all potential differences were checked for. Since users were given a list of potential differences in order of appearance, they knew what to look for; this could have resulted in less forgetting (and thus less re-scrolling) than might otherwise happen.

In 2D, the two analyses were nearly horizontally aligned, so all users used physical navigation to find differences instead of virtual navigation; scrolling was used to go further into the notebook rather than to spot differences. As expected, in 2D users scrolled much less than they did in 1D due to the use of physical navigation and externalized memory on the screen.

| Item | | Rating | | | | | | |
|---|---|---|---|---|---|---|---|---|
| Layout | Question | Strongly Agree | Agree | Agree a little | Neutral | Disagree a little | Disagree | Strongly Disagree |
| **1D** | Easy to Navigate | 4 | 8 | 6 | 1 | 7 | 3 | 1 |
| | Quickly Find Info | 4 | 5 | 7 | 6 | 5 | 2 | 1 |
| | Easy to Compare Graphs | 3 | 4 | 5 | 1 | 5 | 10 | 2 |
| | Easy to Compare Numbers | 1 | 7 | 3 | 0 | 8 | 7 | 4 |
| | Easy to Compare Code | 0 | 4 | 5 | 1 | 5 | 8 | 7 |
| **2D** | Easy to Navigate | 18 | 8 | 4 | 0 | 0 | 0 | 0 |
| | Quickly Find Info | 16 | 13 | 0 | 0 | 0 | 0 | 1 |
| | Easy to Compare Graphs | 19 | 9 | 2 | 0 | 0 | 0 | 0 |
| | Easy to Compare Numbers | 19 | 6 | 4 | 0 | 0 | 1 | 0 |
| | Easy to Compare Code | 24 | 4 | 0 | 1 | 0 | 1 | 0 |

Figure 5: A heatmap comparing the ratings for the Post-1D and Post-2D questions.

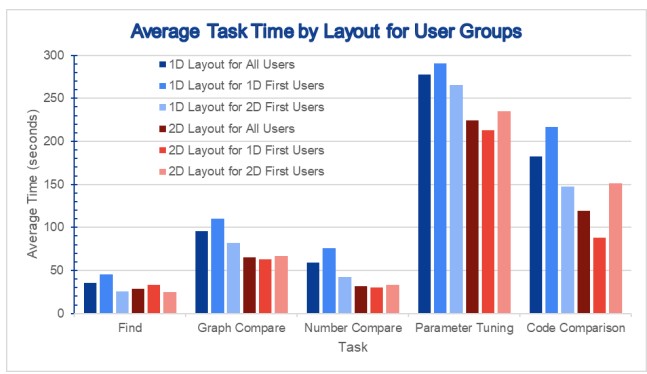

Figure 6: A bar chart showing average time to completion by task and layout in seconds.

## 5.2 Efficiency Measurements

As seen in Table 1 and summarized in Figure 6, we found the layout (1D or 2D) was statistically significant for all tasks except the find task in both 2-Factor ANOVA and in the Scheirer-Ray-Hare. The lack of significance for the find task may be due to it being a "cold find", one without prior knowledge of the notebook, which fails to make use of the benefits of Space to Think. The interaction between layout and order (1D First or 2D First) was significant for the code comparison task.

Analysis of mean and median differences showed the 2D layout resulted in statistically significant improvements to efficiency, summarized in Table 1; these improvements ranged from about 20-50% time reduction. These results likely reflect faster navigation of numerous code cells during the data science tasks when the cells are organized into columns.

### 5.2.1 Accuracy Measurements

The accuracy for 2D and 1D, measured in the number of correct answers given across all participants, was similar for 1D and 2D. 96% of questions in 1D were answered correctly, compared to 98% of questions in 2D.

## 5.3 Survey Questions

We divide the survey question results into 3 areas: Post-1D & Post-2D Questions, Post-Tasks Questions, and Qualitative Comments.

### 5.3.1 Post-1D & Post-2D Questions

As seen in the bar chart in Figure 7, the heatmap in Figure 5 and the results of Table 2, the user impressions of the usability of 2D layouts were significantly more positive than the 1D layouts on all metrics. Users rated 2D approximately 2-4 points higher (on a 7-point likert

Table 2: Post-2D minus Post-1D Average Differences in Rating

| Question | Mean | Median |
|---|---|---|
| Easy to Navigate | **1.87** | **2.00** |
| Quickly Find Info | **1.80** | **2.00** |
| Easy to Compare Graphs | **2.87** | **3.00** |
| Easy to Compare Numbers | **2.83** | **3.00** |
| Easy to Compare Code | **3.57** | **4.00** |

Bolded values are statistically significant with a 0.05 threshold for both paired t-test and Wilcoxon. Positive values indicate 2D is considered better.

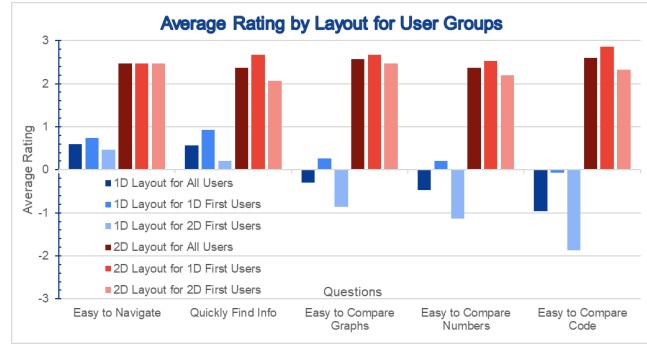

Figure 7: A bar chart comparing the mean ratings for the Post-1D and Post-2D questions; positive values indicate agreement with the sentiment, while negative values indicate disagreement.

scale) than 1D on each metric. Users were nearly unanimously positive in rating 2D, and more evenly divided between positive and negative for 1D. Two participants gave the three negative ratings for 2D in Figure 5; one saw clutter in 2D notebooks as a potential issue, and the other felt the 2D notebooks could be improved by snapping cells next to each to ensure proper alignment of related cells.

Interestingly, as seen in Figure 7, participants exposed to 2D before 1D rated the usability of 1D as significantly worse for the usability questions asked. Thus, exposure to the 2D layout makes the 1D layout seem less usable.

### 5.3.2 Post-Experiment Questions

As seen in the Figure 8 heatmap, when explicitly asked to compare their experiences with the two layouts, participants overwhelmingly viewed 2D as more effective for common data science tasks, especially comparisons, and felt the 2D layout improved their performance. They also agreed that 2D made better use of screen space, and that this was key to their success. Furthermore, most participants seemed interested in using 2D layouts instead of 1D layouts, with only one participant expressing neutrality.

| Item | | Rating | | | | | | |
|---|---|---|---|---|---|---|---|---|
| Category | Question | Strongly Agree | Agree | Agree a little | Neutral | Disagree a little | Disagree | Strongly Disagree |
| 2D Better than 1D at <task> | Navigate | 15 | 7 | 2 | 4 | 2 | 0 | 0 |
| | Locate Items | 17 | 8 | 4 | 1 | 0 | 0 | 0 |
| | Organize & Clean | 10 | 10 | 5 | 3 | 2 | 0 | 0 |
| | Present | 9 | 7 | 3 | 7 | 3 | 1 | 0 |
| | Explore & Prep Data | 12 | 12 | 5 | 0 | 1 | 0 | 0 |
| | Analyze & Develop | 14 | 9 | 1 | 5 | 1 | 0 | 0 |
| | Debug code | 12 | 11 | 4 | 1 | 2 | 0 | 0 |
| | Compare results | 27 | 3 | 0 | 0 | 0 | 0 | 0 |
| | Collaborate | 8 | 8 | 3 | 9 | 2 | 0 | 0 |
| Statements about 2D Layout | 2D Spatial Layout Improved Performance | 19 | 7 | 4 | 0 | 0 | 0 | 0 |
| | More Cells on Screen in 2D Improved Performance | 18 | 8 | 3 | 0 | 1 | 0 | 0 |
| | 2D Layouts Better Used Screen Space | 21 | 6 | 3 | 0 | 0 | 0 | 0 |
| | Would Use 2D instead of 1D | 17 | 10 | 2 | 1 | 0 | 0 | 0 |

Figure 8: A heatmap visualizing the ratings for the Post-Tasks questions.

Table 3: Qualitative Themes in Study 1 Survey

| Theme | Sample Quote | Count |
|---|---|---|
| **Positive Comments on 2D** | | **20** |
| Better Comparison in 2D | "[2D] seems like a solid choice for a lot of analysis applications where you want to do similar but slightly different processes and compare the results." | 7 |
| Better Navigation in 2D | "It was more intuitive and easier to compare side-by-side sections compared to having to scroll so much. I spent so much time scrolling in [1D] that I forgot what I had looked at previously." | 6 |
| Practice with 2D Would Help Improve Performance | "This was my first experience with 2D notebooks after extensive use of 1D notebooks, so the advantages would be compounded given more time to familiarize myself." | 3 |
| 2D is Better Than 1D | "There is no reason anybody should be using 1D anymore." | 2 |
| Other | "We always have to run multiple iteration with different parameter to calculate results and so 2D makes it very easy to see our progress in the notebook and also can be easily inferred." | 2 |
| **Thoughtful Feedback on 2D** | | **6** |
| Column Width & Amount | "Putting too many columns in one screen caused [confusion] and potentially [increased scrolling]." | 2 |
| Arrow Key Navigation | "I found the 2D notebooks were more quick to navigate, but it was easier to navigate the 1D notebook using keys rather than the mouse, which might have been a little bit faster." | 1 |
| Cluttering Screen Space | "I believe one of the only things I might do in a 2D notebook that wouldn't be as easy would be displaying some visuals, as the layout would make them smaller, along with the text. Also having two visuals right next to each other might be seen as cluttered." | 1 |
| Use with Lower Resolutions | "The 2D notebooks were definitely easier to use, but for some tasks/cases (such as presenting on a [low-resolution monitor], or collaborating with... a low-resolution monitor) that might change." | 1 |
| Setup Time | "The only downside I could see is it taking slightly more time to initially set up..." | 1 |
| **Skepticism about 2D** | | **2** |
| Presentation Skepticism | "[1D] looks more clean if you were to present something to another person." | 1 |
| Debugging & Dev Skepticism | "For development and collaboration the linear 1d notebook would be easier to debug." | 1 |

One curious result is that participants expressed skepticism about 2D layouts being better for presenting computational narratives and collaborating with others. Harden et al. [13] found the opposite; debugging, analysis and development, and navigation were seen as weaknesses of 2D layouts, while presentation and collaboration were seen as strengths. This difference may be due to the tasks that users performed in each study; presentation was key for Harden et al. [13], whereas debugging and comparison were key in this study.

### 5.3.3 Qualitative Comments

Of the 27 participants who left a qualitative comment on the survey, 20 expressed positivity about the 2D multi-column layout, while only 2 expressed that they might still prefer 1D notebooks for any task. 2 participants went so far as to express sentiments suggesting that the multi-column 2D layout makes 1D obsolete. 6 participants also left thoughtful feedback that may inform design of future 2D computational notebooks. Several comments pointed out the link between memory and navigation, that more time scrolling in 1D led to more forgetting important information for the task. The results are summarized in Table 3 with all of the themes found, a sample quote for each sub-theme, and the number of comments matching the theme.

Table 4: Scroll Event Analysis Totals Across All Participants

| Measure | 1D Layout | 2D Layout |
|---|---|---|
| Sum of Scroll Event Times | 2071 seconds | 561 seconds |
| Count of Scroll Events | 410 events | 195 events |
| Mean Time per Scroll Event | 5.05 seconds | 2.88 seconds |
| Median Time per Scroll Event | 4 seconds | 2 seconds |

### 5.4 Scrolling Time

For the code comparison task, we found participants scrolled more times and spent more time scrolling in 1D, as seen in Table 4; the average scroll event in 1D tended to be longer than those in 2D, as well. Given differences in typical user interactions described earlier, specifically the elimination of the need to scroll and reduction of scrolling distances for comparison, it makes sense that the 2D layout would have much less scrolling time and events for the Code Comparison task. This confirms that reducing scroll navigation is an important factor in enabling the faster performance results of 2D. This may be due to multi-columns bringing cells nearer to each other

and fitting more cells on the screen simultaneously.

# 6 STUDY 2 METHODOLOGY

The second study focused on understanding how users would utilize the 2D space when starting nearly from scratch, as well as evaluating the longitudinal usability of the 2D Jupyter extension for writing code in a more ecologically valid setting. It consisted of a main task, interview, and a survey.

## 6.1 Recruitment

Participants were recruited from undergraduate and graduate computer science classes at a large state university, and were invited to participate if they had prior experience using Python and computational notebooks. In total, 9 participants completed the study.

## 6.2 Hardware Used in Study

For this study, participants used their personal computers to complete the task. Most participants conducted the task on a laptop using the laptop display and built-in trackpad. Two participants connected their laptops to a 64-inch 4K monitor and extended their displays to the larger screen, but continued to use the laptop's built-in trackpad for navigation and scrolling.

## 6.3 Task Design

A data analysis task for this study was designed that would allow participants to utilize all of the 2D Jupyter extension features. Participants were given a Jupyter notebook file containing task instructions, initial library imports, and loading of two datasets: a COVID dataset containing the number of cases and deaths in each county in the US, and a demographics dataset containing the population of each US county as of the 2020 census.

### 6.3.1 Original Task

The first five participants were instructed to use 2D Jupyter to compare the deaths from COVID among counties in Virginia. Additionally, they were asked to evaluate the correlation of deaths in each county of Virginia with the population density of the county.

### 6.3.2 Modified Task

For the remaining four participants, the data analysis task was modified to introduce more complexity and encourage more flexibility in the use of the 2D environment. These participants were asked to analyze COVID data for three states: Virginia, Texas and Illinois. For each state, they were instructed to create a series of charts showing the top 10 counties in each state with the highest cases, deaths, and deaths per case. Additionally, they were asked to make charts showing the correlation of COVID case and death numbers with the population in the counties in each of the states with the highest number of cases and deaths. Participants were also required to answer a series of questions about the charts they created and make comparisons between the charts.

For all participants, an initial meeting was scheduled to give an overview of the 2D Jupyter extension and to go over the data analysis task. Each participant was allowed to complete the task at their own pace over the course of 2 weeks. An interview session was scheduled after each participant had completed the task, in which they were asked questions about their experiences using 2D Jupyter. At the end of the interview, each person was asked to complete a survey.

## 6.4 Interview and Survey Questions

After completing the data analysis tasks, participants were interviewed about their experience using 2D Jupyter. Interview questions were focused on understanding how the participant used the 2D layout and what features they utilized. Additionally, the participant was asked to share their opinion on any advantages or disadvantages

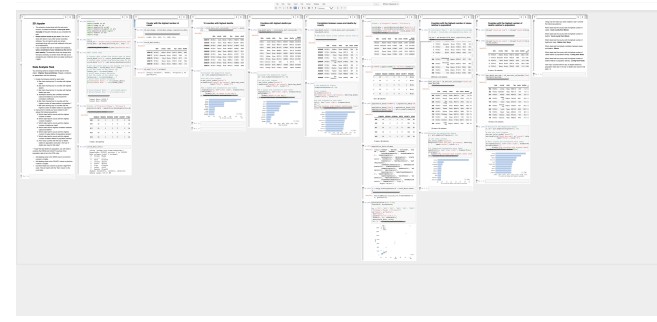

Figure 9: Participant notebook created in Study 2

that 2D notebooks had as compared to traditional 1D notebooks. The questions asked during the interview included:

1. What was your overall strategy for using the 2D environment?

2. What features of the 2D notebook did you utilize?

3. Are there any features that you wish you had?

4. Were there any difficulties in using the 2D notebook during your data analysis?

5. Did the 2D environment provide any advantages for this task as compared to a 1D notebook?

6. Did the 2D environment provide any disadvantages for this task as compared to a 1D notebook?

A survey was also given to participants after the interview session; it consisted mainly of Likert-scale questions. The first four questions of the survey focused on the benefits of a 2D layout in completing the main parts of a data analysis task. The next three questions evaluated the usability of 2D Jupyter. Finally, the survey included two short-answer questions to allow users to provide any suggestions and comments they had regarding their overall experience.

## 6.5 Data Analysis Process

The interview and survey responses were analyzed for themes using open coding by the second author. For strategies for using the 2D environment, the second author also checked the participant notebooks to verify. After the initial pass, feedback from the other authors was sought on the themes and used to refine them. Then, a second pass was made with quotes grouped into themes and organized by the questions seen in Section 6.4.

# 7 STUDY 2 RESULTS

Results of this study are primarily qualitative. A summary of these results can be found in Table 5 with the common themes, a sample quote for each theme, and the number of participants who made comments matching the theme.

## 7.1 Strategies for Using the 2D Environment

For the original data analysis task, we found two main strategies for using 2D space. The first strategy, which 3 participants used, was to use a separate column for each question they were asked to answer. Each column contained the entirety of the analysis needed to answer the question, with the exception of one participant who used two columns to answer the second question to reduce the amount of vertical scrolling needed to view the entire notebook. The second strategy, used by one participant, was to use the columns to separate the steps of the data science workflow, such as data pre-processing,

Table 5: Qualitative Themes in Study 2 Interviews and Survey

| Theme | Sample Quote | Count |
|---|---|---|
| **Advantages of 2D** | | **5** |
| Better navigation | "It was easier for me to find the exact cell that I was looking for." | 2 |
| Better organization | "I don't know that there's any extra challenges from [2D]...I think it's strictly better organizationally" | 1 |
| Ease of comparisons | "...when I have to compare two data frames...side by side that's really useful." | 2 |
| **Disadvantages of 2D** | | **2** |
| Viewing on Small Screens | "...the major disadvantage is all the [horizontal] scrolling that you have to do." | 1 |
| Cluttered Look | "..it can look kind of cluttered sometimes, like it can be maybe a little overwhelming..." | 1 |
| **Usability Feedback** | | **8** |
| Column Resizing | "...if it was possible to resize it directly from [the middle of the column] instead of having to go up and resize, that would be good.." | 2 |
| Column Scrolling | "I would like each columns to have their own [independent] scrolling area" | 1 |
| Easy to Learn | "...after that that small little learning curve, I think everything else was...super straightforward" | 2 |
| Opportunities | "I don't see there being like any sort of disadvantage or any type of limitation that 2D has compared to 1D. If anything...the opportunities are endless." | 3 |

Table 6: Number of Columns Used by Participants

| Number of Columns Used | Number of Participants |
|---|---|
| 1 | 2 |
| 2 | 1 |
| 3 | 2 |
| 4 | 2 |
| 6 | 1 (4k screen) |
| 10 | 1 (4k screen) |

data exploration, and so on. Each column was treated as a new section of the overall notebook.

For the modified data analysis task, each participant had a different strategy for using the 2D space. One participant used the columns as sections of their notebook, creating a new column when they began working on a new data science subtask. Another participant used columns to reduce scrolling and only created a new column when the vertical length of the page became too long. One participant used a single column of cells alongside a single markdown scratch cell containing the task instructions; they used the freeform cell placement ability to move the markdown cell down the page as the page became longer. The last participant created only two columns and placed cells side by side when they wanted to reference code they wanted to reuse or to compare visualisations.

Finally, one participant used 2D Jupyter for their own project rather than the given data analysis task. This participant was a student in an artificial intelligence class and was working on a project to build their own AI model that could play a game. This participant primarily used the freeform scratch cell feature of the extension to test parameters for their model, rather than using multiple columns.

Table 6 shows the number of columns used by participants. 2 participants created 1 column of cells alongside a freeform scratch cell that they moved around the notebook outside the column as they worked. One participant used 2 columns, primarily using the second column to place cells side-by-side for referencing code or comparing visualizations. Most participants used 3-4 columns to complete the data analysis task. The 2 participants who used the large 4K display created the most columns, using 6 and 10 columns in their completed notebooks.

### 7.2 Advantages of 2D Over 1D

During the study, participants were asked to identify areas where they felt 2D notebooks had an advantage over 1D notebooks. Participant responses are summarized below, and participant comments can be found in Table 5.

First, referencing other cells is found to be easier in 2D. 5 participants liked that the 2D environment made it easier to refer to cells for reusing code or comparing charts and visualizations. The ability to place cells side-by-side using multiple columns or by placing a scratch cell next to a column reduced the amount of scrolling required while conducting the data analysis and did not disrupt organization of the notebook.

Additionally, 2D notebooks made it easier to locate specific code or data. 3 participants noted that the 2D environment made it easier to keep track of cells due to the organizational benefits of the 2D layouts. With the multi-column structure, as long as the section that the cell belonged to was known, users could physically navigate to find the relevant section a cell was in and more quickly find the cell instead of having to scroll to find the section.

Finally, 2 participants liked that they were able to view more of their code at once. A typical laptop display allows for 3-4 columns on the screen at once, resulting in 3-4 times more code cells viewable in the display. Larger displays can fit even more columns on the screen at once, thus increasing the number of cells viewable on the screen. One participant also felt that the 2D environment provided a better organizational structure.

### 7.3 Disadvantages of 2D Compared to 1D

Participants also identified several areas where 2D had a disadvantage over 1D notebook.

One main disadvantage found is that extra horizontal scrolling is required to navigate, especially on smaller screens. One participant noted that while vertical scrolling was reduced, the smaller screen size of typical laptops would require more horizontal scrolling in order to view the entire notebook.

Additionally, participants suggested that the 2D environment can make it harder to find a "lost" cell. In other words, if a user forgets which section they have placed a cell that they want to revisit, they would now have to scroll both vertically and horizontally to search for the cell. In a 1D notebook, users could scroll in just the vertical direction until they found the cell they were looking for. A 2D search pattern is more complex than a 1D search pattern, and may lengthen the search time.

Finally, presenting a 2D notebook is more challenging. Two participants felt that presenting a notebook created in the 2D Jupyter environment would be harder than a 1D notebook. One participant commented that navigation through two dimensions made it harder to read the entirety of the notebook, and it could potentially look cluttered if there are too many columns and cells. Additionally, the extension does not currently support exporting the notebook layout to another file type such as a HTML or PDF format, making it harder to share the notebook.

Notably, three participants did not identify any disadvantages to the 2D Jupyter environment. These participants suggested that at

| Statement | Strongly Agree (2) | Agree (1) | Neither agree nor disagree (0) | Disagree (-1) | Strongly disagree (-2) |
|---|---|---|---|---|---|
| I found the use of the 2D extension to be beneficial in organizing my notebook. | 2 | 3 | 1 | 0 | 1 |
| I found the use of the 2D extension to be beneficial in data processing. | 1 | 2 | 3 | 0 | 1 |
| I found the use of the 2D extension to be beneficial in creating visualizations. | 3 | 2 | 1 | 0 | 1 |
| I found the use of the 2D extension to be beneficial in debugging my code. | 0 | 4 | 2 | 0 | 1 |
| It was easy to understand how to use the 2D extension. | 2 | 5 | 0 | 0 | 0 |
| It was easy to navigate in the 2D notebook. | 0 | 5 | 1 | 1 | 0 |
| I prefer using the 2D notebook environment over the traditional notebook environment. | 0 | 3 | 4 | 0 | 0 |

Figure 10: Survey results from Study 2

most, there would be a small learning curve as users got used to a new layout, but otherwise the 2D environment was not taking away from the 1D notebook environment.

### 7.4 Suggestions and Improvements

Participants had the opportunity throughout the study to provide comments and suggestions on the 2D extension. Primarily, participants wanted shortcut access to the new toolbar controls. For example, multiple participants wanted to add code cells from anywhere in the notebook, without needing to use the toolbars at the top of the columns. Other participants wanted to resize the columns without needing to scroll to the top of the column to find the resize controller. Two participants suggested adding the ability to independently vertically scroll through a column while keeping the rest of the notebook static. One participant wanted to be able to concurrently run cells placed side-by-side without having to run each cell individually.

Several participants found that orienting themselves in 2D space was somewhat challenging and provided suggestions for improvement. One participant suggested adding a mini-map at the bottom corner of the screen to see their location within the overall notebook. Other participants suggested labelling each cell with a row and a column number, similar to how Excel spreadsheet cells are labeled.

### 7.5 Survey Questions

All 9 participants in Study 2 were asked to complete the survey, but 2 participants skipped the questions in their responses, resulting in 7 total responses. The results of this survey are shown in Figure 10.

The heatmap shows participants generally viewed 2D notebooks positively. When asked if the 2D layout was beneficial in completing common data analysis tasks, most participants agreed or strongly agreed with the statements. In terms of usability, all participants agreed with the statement that it was easy to understand how to use the 2D extension. Additionally, most participants agreed that it was easy to navigate in the 2D layout. When asked if they would prefer using the 2D extension over the traditional 1D environment, all participants were either netural or agreed with the statement.

## 8 Discussion

We divide our discussion into the following categories: task efficiency benefits, usability benefits, effects of hardware, design challenges and opportunities, and limitations of our work.

### 8.1 Task Efficiency Benefits

The multi-column 2D computational notebook layout provides benefits to task efficiency by reducing the amount of scrolling necessary and shortening the length of needed scrolls. As seen in Study 1, the multi-column layout provided statistically significant reductions in time to completion overall. Given how much less scrolling was done in terms of total scrolling time, number of scrolling events, and average scrolling time in the multi-column layout, per Study 1's Scrolling Time analysis, combined with the time to completion results, the multi-column layout clearly provide benefits to efficiency.

The reduced scrolling is a result of 2D's ability to bring more cells nearer to each other. Theoretically, 2D can reduce distances by the square root of 1D distances. Practically, 2D enables non-linear code structures, such as parallel analyses, to be horizontally aligned in columns, thus supporting common data-science tasks such as comparison. 2D enabled more such relationships to be encoded into the space. In contrast, 1D encodes only a single ordering, and would require complex refactoring tools to enable various types of parallel analyses and comparisons.

### 8.2 Usability Benefits

The multi-column layout appear more usable for certain basic and more complex tasks. Based on the results from Study 1 as seen in Figures 5 and 8, navigating and finding information, comparing results, and data science tasks such as organizing and cleaning may be easier in a multi-column notebook. This may be due to more effective use of screen space to display more information at once in an organized manner, along with more efficient scrolling options.

In Study 2, several participants found the 2D, mainly multi-column environment provided an advantage in locating code or data. The ability to break up the notebook into distinct sections meant they did not have to first search for a section of their notebook, and then search for the info they needed within the section; the multi-column layout enabled users to more easily find the info they were looking for, since they could instantly identify the section of the notebook they needed.

Additionally, participants found that the 2D, mainly multi-column environment made it more convenient to refer to other cells. In the 1D environment, users would need to move two cells close to each other in order to easily compare the contents, often disrupting the organization of the notebook. In the 2D environment, participants were able to maintain the organization of the cells in their respective sections, while still being able to place cells next to each other for ease of comparison.

### 8.3 Effects of Hardware on 2D Computational Notebooks

Different setups, especially in the second study, made for different experiences with 2D Jupyter. Specifically, both screen size and scrolling device (e.g. trackpad vs. mouse) affected usability. Larger screen sizes afforded the ability to visualize more columns at once

and to better ensure those columns were sufficiently wide for the code. This enabled physical navigation more effectively than smaller screens and thus led to less scrolling. Furthermore, scrolling devices which easily enable horizontal scrolling through simple gestures, such as trackpads, appeared to provide a better user experience with the 2D layout than a standard mouse or vertical scroll wheel. Without an easy way to scroll horizontally, like for some Study 2 participants, such scrolling may become more tedious and costly.

There is a tradeoff between vertical and horizontal scrolling. Using more columns reduces vertical scrolling, but increases horizontal scrolling. On small displays, some users in Study 2 indicated that intensive use of both types of scrolling may be worse than just vertical scrolling. However, large widescreen displays, increasingly common in data-science workspaces, mitigate this tradeoff by minimizing the horizontal scrolling needed to traverse the notebook, enabling multiple columns to greatly reduce vertical scrolling. Even with a modest 24" display, like in Study 1, the benefit was significant, and would likely increase with larger displays.

### 8.4 Design Challenges & Opportunities

While multi-column 2D computational notebooks may provide efficiency and usability benefits, especially with the right setup, there is still room for improvement on their design, especially as it relates to managing column width and navigating the notebook.

Column width in the multi-column design pattern may impact user experience; if the columns are too wide, fewer columns will fit on the screen, but if the columns are too narrow, visuals may become too small to easily read and the screen may feel cluttered, potentially leading to confusions that affect performance. Thus, managing column width becomes an important factor; this is currently doable in 2D Jupyter through manual resizing of columns. Still, it may be beneficial to provide functionality that resizes columns to an ideal width through a quick interaction, such as is done in spreadsheets.

Additional navigation options tailored to different 2D layouts may also benefit users. Navigating 1D computational notebooks with arrow keys can be quicker than navigating with manual scrolls, and the same may apply to 2D computational notebooks; the challenge is whether and how to incorporate the left and right arrow keys (or even diagonals) to quickly navigate. One option is to borrow the grid metaphor and have each arrow key move to the adjacent cell in the direction of the key. Making individual columns independently scrollable may also benefit navigation, especially when working on smaller screens. This would allow longer columns to be scrolled without impacting the view of shorter columns.

### 8.5 Limitations

Our work has some limitations, especially as it concerns bugs in our extension and lack of sufficient support for 2D organizational patterns other than multi-column at the time of the studies.

#### 8.5.1 Bugs in Extension

At the time of conducting both studies, the 2D Jupyter extension contained some bugs that could affect user experience. In particular, the drag and drop feature occasionally did not allow the user to release the cell at an intended location, forcing the user to reload the page. Additionally, the layout of the 2D environment was sometimes not properly saved between kernel sessions, requiring the user to reorganize their notebook before resuming work; the extension required users to manually save their work as the autosave feature built into Jupyter Notebooks did not work with the extension. These bugs did not affect Study 1 except for contributing to the technical issues that led to discarding one participant's data.

#### 8.5.2 2D Layouts other than Multi-Column

Given that both studies used an extension which does not, at the time of this writing, fully support 2D layouts other than multi-column,

care must be taken in assigning benefits to other 2D layouts. Some of the advantages of the multi-column layout may be due to how compact it is; less compact 2D layouts might not see the same level of benefits in some areas, like reduced scrolling and task efficiency, without specialized navigational tools. Evaluating other 2D layouts is a subject for future work.

## 9 CONCLUSION

Computational notebooks are a potent tool for creating and presenting computational narratives; the 1D layout of notebooks, while elegant in its simplicity, imposes certain limitations that make comparative analyses and navigating longer non-linear notebooks, among other tasks, more difficult. Thus, we developed an extension and evaluated the potential of 2D for computational notebooks, starting with the multi-column layout enabled in our 2D Jupyter extension.

The multi-column 2D layout provides benefits in efficiency and usability for common data science tasks such as comparative analyses by enabling greater physical navigation, thus minimizing the scope and need for virtual navigation (scrolling). In addition, the multi-column layout provides an effective sectioning mechanism that may help combat messiness along with providing more efficient navigation. While our conclusions are limited to the multi-column layout, 2D layouts may improve upon the current state of computational notebooks and provide a novel way to enhance the creation and presentation of non-linear computational narratives through enabling Space to Think.

### ACKNOWLEDGMENTS

This publication is based on work supported in part by the National Science Foundation, awards 2004014, 2003800 and 2003387 for SAGE3, and a Virginia Tech College of Engineering New Horizon Graduate Scholarship.

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
