# OpenReview forum: "“There is no reason anybody should be using 1D anymore”: Design and Evaluation of 2D Jupyter Notebooks"
_graphicsinterface.org/Graphics_Interface/2023/Conference_SD — GI 2023 - second deadline_

### Official Review · Reviewer_SJT3 · 2023-04-13
**good overall but have some concerns**

**Rating:** 5
**Confidence:** 4

**Review:**

This paper presents an empirical study comparing 1D and 2D Jupyter Notebooks on completing data science related tasks. The topic of this paper is very interesting and important. Overall, the paper is motivated well and structured well. However, there still exist several concerns.

First, it is unjustified why this specific 2D Jupyter Notebook design is employed in the study. The 2D Jupyter Notebook is customized with columns that allow cells to be moved around. When extending to a 2D canvas, there are multiple design alternatives. One could be the Figma style that allows cells to be freely placed on the canvas. Another design could be like ForkIt, with some rows containing more than one cell. Or, a grid/matrix like design, with cells constrained in rows and columns. Why comparing the classic 1D Notebook with the specific 2D design in this paper is not well justified. This is the biggest concern of this paper, as the results may not be generalizable to other 2D designs.

Second, there were 62 participants passing the screening and invited to do the study, and there were only 31 participants completing the study. Only 50% of the data was actually used in the results and analysis. Why the task completion rate is so low? Is it due to improper design of the task? Also, even the participants didn’t complete the study task, their feedback and interactions are still valuable to report. Here it needs more explanation on why only half of the data was used.

Third, the metrics in study 1 included completion time and accuracy, but they are not properly defined. The study tasks are pretty exploratory: results comparison, parameter tuning, and code comparison. Due to the fuzziness of the tasks, it is hard to define when is the completion and what is correct. Here needs more clarification on the metrics. To me, the qualitative comments and observation of participants’ behaviors are more valuable; however, this part is very thin in this paper. I hope study 2 could complement on this aspect, but Sec 7.2 and 7.3 are very short, lacking depth.

Minor:
The figures can be improved with high resolution ones, especially Fig 1

---

### Official Review · Reviewer_7qGV · 2023-04-23
**A clear contribution to knowledge about Jupyter Notebooks**

**Rating:** 8
**Confidence:** 4

**Review:**

In this paper, the authors set out to assess the effectiveness of 2D layouts for Jupyter Notebooks when compared to regular notebooks layout (1D).
The paper is overall well written and the analysis, questions, choice of tasks are well motivated throughout the paper. It is an interesting and timely read that I particularly enjoyed. I also appreciated the fact that the authors gave the link to their tool in the paper. My detailed comments (and summary decision about the submission) can be found below.

I particularly appreciate that the authors have given exact p-values for all of their results in a table (for all p > 0.001, which is quite common, although for these small values the authors might want to consider using scientific notations, see [A]). While the effect size (expressed in % in table 1) is interesting, I did not find that the authors discussed it enough in the manuscript. But that is a rather minor point. That being said, I have a couple of questions about the statistical analysis or the presentation of results:
- the authors used ANOVA to analyse their completion time but I wonder whether they have checked that they can use the statistical test that they have used since completion time data is log-normal (and not normal) and therefore might violate statistical assumptions.
- Figure 5 and 6 (and later 7) are tables with a heatmap. This is a good choice as it really highlights differences between the 2 layouts. But the tables use green and red which might not be easily distinguishable by colour-blind people. I would suggest adjusting the colour scales to take this into account.
- Table 3 is quite informative and an interesting way to summarise the findings of the authors. Is the whole analysis summarised in this table? This did not appear quite clearly to me through the text in 5.3.3. What I mean by this is: is there any other theme not mentioned in this table? I would also suggest trying to save some space with this table which is taking more than half a page by reducing by half (or so) the last column such that the quotes would take fewer lines (perhaps some quotes could also be modified to be eventually slightly shorter).

Since the authors have released their code, I would argue that they could probably go the extra mile and release their data, questionnaires, and data-analysis code to be even more transparent and provide reusable research materials.

I have to admit that I was slightly puzzled to see any discussions of Jupyter Notebooks and their advantages (and limitations) to foster more transparent research papers and data analysis. I would argue that the paper would benefit from anchoring Jupyter Notebooks in the framework of practices and ideas to provide more transparent research manuscripts and avoid replication issues. In particular, I can think of approaches such as [B] which aimed at providing transparent and interactive multiverse analysis in research outputs in order to foster more transparency and avoid replication issues [A]. I can see Jupyter Notebooks as a clear advances in this direction, or perhaps as a complimentary approach and I would argue that the authors would strengthen their manuscript and motivate their work further by discussing these and anchoring their motivation also in this aspect. A dedicated subsection of their related work could for instance provide such discussions and they could also try to include them in their discussions. In fact the argument that Jupyter Notebooks can be used to increase reproducibility has been made in other papers (e.g., [C]), and the issue of non-reproducible research is pervasive across research domains and does not evade HCI [D] or empirical computer science [A]. Other relevant papers include [E,F,G] and further advance the benefits of Jupyter Notebooks for reproducible science.

Overall, this is an interesting paper with a robust and well motivated set of experiments that advances both of our understanding of Jupyter Notebooks and provides a way to improve them through a developed extension. I would be in favour of accepting the submission, provide that the authors can address the very minor changes I have listed:
- redoing their heatmap figures
- justifying the assumptions for their statistical tests (some text to add)
- anchoring the importance of Notebooks for reproducible science (a subsection in the related work and/or discussions at the end of the paper)


References:
- [A] http://dx.doi.org/10.1145/3360311
- [B] https://dl.acm.org/doi/abs/10.1145/3290605.3300295
- [C] https://arxiv.org/pdf/1810.08055.pdf
- [D] https://dl.acm.org/doi/10.1145/3290607.3310432
- [E] https://doi.org/10.1109/MSR.2019.00077
- [F] https://doi.org/10.1145/3324884.3416585
- [G] https://doi.org/10.1109/MCSE.2021.3052101

---

### Official Review · Reviewer_pz5A · 2023-04-24
**Simple but interesting extension of 1D notebooks, but paper needs some targeted fixes.**

**Rating:** 7
**Confidence:** 4

**Review:**

This submission presents a straightforward but interesting extension of standard linear data analysis notebooks that allows users to break notebooks into multiple columns. The authors present two studies that examined participants' experiences using existing multi-column notebooks and creating their own, and highlight some positive initial impressions from both.

Overall, I think the paper presents a small but interesting idea and grounds it in a pair of studies that provide some evidence of the utility of the approach. I can imagine a number of places in which a multi-column layout might indeed be useful, and I think the work is likely to be of interest to some users and developers of Jupyter and related platforms.

I do have some concerns about the current submission, however I think most of these could be resolved in a light revision:

* **Claims about the effectiveness of 2D notebooks are too strong.** The paper makes a bunch of claims about 2D notebooks generally that seem unjustified given that the authors have really only examined one simple multi-column approach. While the authors give a brief disclaimer to this effect in 8.5.2, I would prefer to see the claims throughout the paper softened to focus on just this multi-column approach rather than 2D layouts generally. Talking only about the "multi-column" layouts rather than more general "2D" ones might the most appropriate, since outside of Section 3, other layouts (like the "workboard" style mentioned there) are not really considered. The reduced scrolling seen in this specific 2D column-based format also seems likely to be highly dependent on the layout of the notebooks. Even some layouts using the current column structure — say one long column surrounded by a series of much shorter ones — could potentially make it *much easier* for users to become lost or lead to excessive scrolling. With that in mind, I think it would be best to offer some qualifications when discussing scrolling.

* **The paper needs to clarify the relationship between layout and execution order.** Confusion about the execution order of cells is a very common issue in 1D notebooks, and seems likely to be an even bigger problem in 2D. However, the paper never mentions execution order or clarifies how the 2D layouts interact with it. By default does execution order run down the first column, to the second, then third, etc.? Did participants have any issues with execution order?

* **Images of the system are basically illegible.** The system itself is really only shown in the teaser image, and then at a small size and low resolution that mean it's essentially impossible to see what is happening or build any intuition about the system. This also makes the references to this figure (from all the way in Section 4 no less) extremely frustrating, since it's hard to see any detail about the analysis notebooks in this image, and definitely not enough to understand how the examples leverage columns. Clearer images of the system, with callouts to show details would be extremely helpful. I'd also strongly encourage the authors to consider including full-scale examples of the notebooks as part of their supplemental material.

* **Additional study details needed.** Study 1 would benefit from some additional detail around recruiting, since it's currently somewhat unclear how invites and selection process were handled and what might account for the extremely high attrition rates. Some clarification around whether or not participants were allowed to use search would also be appreciated, since a simple Ctrl+F seems like it might make it equally easy to complete some of these tasks in either layout.

# Miscellaneous Points
* **Github link anonymity?** While links to the project are appreciated, this link seems to break the anonymity of the authors. Consider using an anonymized repository or other mechanism for sharing this kind of content in future double-blind submissions.

* **Figures 5, 6, and 7 are are images rather than actual tables.** (And low-res images at that!) That makes these tables problematic from an accessibility perspective and also makes it impossible for readers to select any text in them.

* **Subheading->Subsubheading jumps.** The paper repeatedly includes pairs of headings (like 8/8.1 and 8.5/8.5.1) immediately after one another with no text in between.

* Including the complete Study 2 task prompt s(which consumes more than half a page) in the paper itself is probably too much. This could easily be summarized or included as supplemental material.

As an aside, I found myself repeatedly wondering whether other simple 1.5D layouts (for example nesting multiple cells horizontally within a single cell in a more traditional linear notebook) might nicely support parameter tuning or comparison in ways that retain some of the benefits of the multi-column layouts. These and other kinds of nested layouts might be interesting to explore in the future.